# Therapeutic Efficacy and Safety of Lenvatinib after Atezolizumab Plus Bevacizumab for Unresectable Hepatocellular Carcinoma

**DOI:** 10.3390/cancers15225406

**Published:** 2023-11-14

**Authors:** Shigeki Yano, Tomokazu Kawaoka, Shintaro Yamasaki, Yusuke Johira, Masanari Kosaka, Yuki Shirane, Ryoichi Miura, Kei Amioka, Kensuke Naruto, Kenji Yamaoka, Yasutoshi Fujii, Shinsuke Uchikawa, Hatsue Fujino, Atsushi Ono, Takashi Nakahara, Eisuke Murakami, Daiki Miki, Masataka Tsuge, Yuji Teraoka, Hirotaka Kouno, Shintaro Takaki, Nami Mori, Keiji Tsuji, Shiro Oka

**Affiliations:** 1Department of Gastroenterology, Graduate School of Biomedical and Health Sciences, Hiroshima University, Hiroshima 734-8551, Japan; yano0319@hiroshima-u.ac.jp (S.Y.); syamasaki@hiroshima-u.ac.jp (S.Y.); jyusuke9@hiroshima-u.ac.jp (Y.J.); m0607k@hiroshima-u.ac.jp (M.K.); yuki0415@hiroshima-u.ac.jp (Y.S.); ryoichim@hiroshima-u.ac.jp (R.M.); amioka@hiroshima-u.ac.jp (K.A.); uzumaki1@hiroshima-u.ac.jp (K.N.); yamaokak@hiroshima-u.ac.jp (K.Y.); fujiiyasu@hiroshima-u.ac.jp (Y.F.); shinuchi@hiroshima-u.ac.jp (S.U.); fujino920@hiroshima-u.ac.jp (H.F.); atsushi-o@hiroshima-u.ac.jp (A.O.); nakahara@hiroshima-u.ac.jp (T.N.); emusuke@hiroshima-u.ac.jp (E.M.); daikimiki@hiroshima-u.ac.jp (D.M.); tsuge@hiroshima-u.ac.jp (M.T.); oka4683@hiroshima-u.ac.jp (S.O.); 2Department of Gastroenterology, National Hospital Organization Kure Medical Center and Chugoku Cancer Center, Hiroshima 737-0023, Japan; teraoka.yuji.ed@mail.hosp.go.jp (Y.T.); kouno.hirotaka.pz@mail.hosp.go.jp (H.K.); 3Department of Gastroenterology, Hiroshima Red Cross Hospital and Atomic-Bomb Survivors Hospital, Hiroshima 730-8619, Japan; takakishiroshima@gmail.com (S.T.); nami7373@star.ocn.ne.jp (N.M.); k_tsuji_@yahoo.co.jp (K.T.)

**Keywords:** immune checkpoint inhibitor, molecular-targeted agent, first-line therapy, second-line therapy

## Abstract

**Simple Summary:**

Immune checkpoint inhibitor therapy has been rapidly developed for the treatment of unresectable hepatocellular carcinoma (HCC). In the IMbrave150 trial, atezolizumab plus bevacizumab was seen as the first-line systemic drug therapy for unresectable HCC because overall survival and progression-free survival were significantly prolonged compared with sorafenib. However, an effective regimen after atezolizumab plus bevacizumab failure has not yet been established. Lenvatinib, on the other hand, also demonstrated good outcomes in unresectable HCC in the REFLECT trial as first-line therapy and is currently positioned as one of the second-line therapies after atezolizumab plus bevacizumab. The aim of this retrospective study was to evaluate the efficacy and safety of lenvatinib after atezolizumab plus bevacizumab for unresectable HCC.

**Abstract:**

A total of 137 HCC patients treated with atezolizumab plus bevacizumab from October 2020 to September 2022 were enrolled. The median overall survival (OS) and progression-free survival (PFS) from the beginning of atezolizumab plus bevacizumab were 21.1 months (range, 18.8 months–not reached) and 10.5 months (range, 8.2–12.1 months), respectively. Fifty patients were diagnosed with progressive disease after atezolizumab plus bevacizumab. Of this group, 24 patients were administered lenvatinib, and the median OS and PFS from the beginning of lenvatinib were 15.3 months (range, 10.5 months–not reached) and 4.0 months (range, 2.5–6.4 months), respectively. The objective response rates based on the response evaluation criteria in solid tumors (RECISTs) criteria version 1.1 and modified RECISTs were 33.3% and 54.2%, respectively. There was no significant difference in the median serum alpha-fetoprotein level between before and after lenvatinib. In the multivariate analysis, Child–Pugh class A (hazard ratio 0.02, 95% confidence interval (CI) 0.02–0.76, *p* = 0.02) and intrahepatic tumor occupancy rate < 50% (hazard ratio < 0.01, 95% CI 0.003–0.35, *p* < 0.01) were the significant factors for OS. There were some frequent adverse events (AEs) in patients treated with lenvatinib such as hypertension, fatigue, anorexia, proteinuria, and so on, but none directly caused death. In conclusion, lenvatinib after atezolizumab plus bevacizumab for unresectable HCC should be considered an effective treatment option.

## 1. Introduction

Primary liver cancer is the sixth most diagnosed cancer worldwide, and hepatocellular carcinoma (HCC) accounts for more than 90% of primary liver cancers [1]. In Japan, the Japan Society of Hepatology (JSH) published its HCC Guidelines 2021 that determine the treatment strategy for HCC, with the treatment algorithm based on tumor number, tumor size, liver function, metastasis, and vascular invasion. Systemic therapy is the cornerstone of management for patients with advanced and unresectable HCC for whom locoregional therapies are not appropriate, such as surgical resection or radiofrequency ablation. To determine the efficacy of systemic therapy, the blood levels of tumor markers such as serum alpha-fetoprotein (AFP) [2] and des-gamma-carboxyprothrombin (DCP) [3] and imaging examinations to evaluate the size of the tumor and microvascular invasion are used [4]. Until 2018, the only systemic therapy for unresectable HCC was sorafenib, a tyrosine kinase inhibitor (TKI) [5], but the results of several phase III trials recently led to the approval of multiple drug therapies [6,7,8,9]. The REFLECT trial was a phase III clinical trial evaluating the noninferiority of lenvatinib to sorafenib in the first-line treatment of unresectable HCC. The primary endpoint of overall survival (OS) showed the noninferiority of lenvatinib, and for the secondary endpoints of progression-free survival (PFS), objective response rate (ORR), and time to progression, lenvatinib was significantly better [6].

Since 2020, atezolizumab plus bevacizumab has been positioned as a first-line systemic drug therapy for unresectable HCC because it significantly prolonged OS and PFS compared with sorafenib in the IMbrave 150 trial [10]. At the American Society of Clinical Oncology—Gastrointestinal (ASCO-GI) 2021, an updated analysis of the IMbrave150 trial reported OS of 19.2 months and PFS of 6.9 months. The establishment of a second-line therapy after atezolizumab plus bevacizumab combination therapy is an urgent issue. After atezolizumab plus bevacizumab, lenvatinib is used as second-line therapy in current clinical practice. However, not much time has passed since the benefit of atezolizumab plus bevacizumab combination therapy was demonstrated, and the development of second-line therapy for unresectable HCC has been conducted in patients who failed first-line sorafenib therapy. Thus, an effective regimen after atezolizumab plus bevacizumab combination therapy has not been established at present.

In this report, the effectiveness of lenvatinib as second-line therapy after atezolizumab plus bevacizumab is demonstrated.

## 2. Materials and Methods

### 2.1. Patients

This study’s flow chart is shown in Figure 1. A total of 137 patients received atezolizumab plus bevacizumab for unresectable HCC between October 2020 and September 2022 at Hiroshima University Hospital and affiliated hospitals. First, their patient records were examined, and the clinical data obtained at the start of atezolizumab plus bevacizumab were collected. In addition, OS and PFS from the beginning of atezolizumab plus bevacizumab were evaluated.

Next, patients were selected to evaluate lenvatinib after atezolizumab plus bevacizumab. Of the 137 patients, 87 (63.5%) were interrupted for the following reasons: 9 (6.6%) patients were diagnosed as having complete response (CR), 11 (8.0%) patients had adverse events (AEs), 10 (7.3%) patients showed decreased performance status, 9 (6.6%) patients had decreased liver function, 7 (5.1%) patients did not wish to be treated, and 5 (3.6%) patients dropped out because of other diseases. Thirty-six (26.3%) patients continued atezolizumab plus bevacizumab. In total, 50 (36.5%) patients were diagnosed as having progressive disease (PD), and of them, 23 (16.8%) were treated by other treatments, including 19 (13.9%) with transcatheter arterial chemo embolization, 2 (1.5%) with radiation, and 1 (0.7%) with ramucirumab, and 1 (0.7%) had peritoneal seeding. Three (2.2%) patients opted for best supportive care. A total of 24 (17.5%) patients treated with lenvatinib after atezolizumab plus bevacizumab were evaluated. Clinical characteristics and therapeutic response, including OS, PFS, and the ORR, univariate and multivariate analyses of lenvatinib for OS, and AEs were analyzed retrospectively.

### 2.2. Treatment Regimens

Regarding atezolizumab plus bevacizumab, patients received 1200 mg of atezolizumab plus 15 mg of bevacizumab per kilogram of body weight intravenously every 3 weeks. For lenvatinib, patients received full dose (12 mg/day for body weight ≥60 kg, 8 mg/day for body weight <60 kg) until the withdrawal of consent, death, disease progression, worsening of liver function, or unacceptable toxicity.

### 2.3. Efficacy Assessment

Patients’ response to treatment was evaluated every 1–3 months using dynamic computed tomography (CT) or magnetic resonance imaging (MRI), as well as serum AFP and DCP levels. A hepatologist and a radiologist assessed the treatment response according to the response evaluation criteria in solid tumors (RECISTs) criteria version 1.1 and modified RECISTs (mRECISTs) [11] using the following four response categories: CR, partial response (PR), stable disease (SD), and PD. The ORR was calculated as the sum of the patients who attained CR and those who attained PR, and the disease control rate (DCR) was calculated as the sum of the ORR and SD. The calculations of the ORR, the DCR, and the ORR used the best response recorded from the beginning of treatment to disease progression or recurrence. The National Cancer Institute Common Terminology Criteria for Adverse Events version 5.0 was used for the assessment of treatment-related AEs.

### 2.4. Statistics

OS and PFS were calculated using the Kaplan–Meier method and analyzed via the log-rank test. To determine OS and PFS, the dates of treatment initiation to the date of the last follow-up or death and the date of recurrence, respectively, were used. The factors contributing to OS and PFS were identified using a Cox proportional hazards model. A *p*-value < 0.05 was taken as indicating a significant result. All statistical analyses were performed using EZR (Saitama Medical Center, Jichi Medical University, Saitama, Japan), a graphical user interface for R (The R Foundation for Computing, version 3.4.1).

## 3. Results

### 3.1. Patients’ Background Characteristics and Outcomes from the Beginning of Atezolizumab Plus Bevacizumab

The patients’ background characteristics at the beginning of atezolizumab plus bevacizumab are shown in Table 1. The median age was 75 years (range, 47–92 years); there were 107 male patients and 30 female patients. The etiologies of liver cirrhosis were hepatitis B virus in 15 (10.9%), hepatitis C virus in 48 (35.0%), hepatitis B virus and hepatitis C virus in 2 (1.5%), and non-B, non-C viral in 72 (52.6%) patients. The Child–Pugh scores were 5 and 6 in 84 (61.3%) and 53 (38.7%) patients, respectively. Relative tumor volumes <50% and ≥50% were seen in 130 (94.9%) and 7 (5.1%) patients, respectively. The median size of the liver tumor was 28 mm (range, 0–220 mm). The HCC stages were II, III, IVa, and IVb in 33 (24.1%), 50 (36.5%), 22 (16.1%), and 32 (23.4%) patients, respectively. The Barcelona Clinic Liver Cancer (BCLC) stages were A, B, and C in 8, 63, and 66 patients, respectively. The median AFP was 18.1 ng/mL (range, 1.2–63,642 ng/mL). The median DCP was 236 mAU/mL (range, 11–197,680 mAU/mL). The median observation period of atezolizumab plus bevacizumab was 11.7 months (range, 1–28 months).

Figure 2 shows OS and PFS from the beginning of atezolizumab plus bevacizumab. The median survival time (MST) was 21.1 months (95% confidence interval [CI], 18.8 months–not reached). The median PFS was 10.5 months (95% CI, 8.2–12.1 months).

### 3.2. Patients’ Clinical Data and Outcomes after Progression of Atezolizumab Plus Bevacizumab

Patients’ clinical data at the time of the diagnosis of progressive disease regarding atezolizumab plus bevacizumab are shown in Table 2. Systemic therapies after atezolizumab plus bevacizumab were lenvatinib (*n* = 24) and ramucirumab (*n* = 1).

Figure 3 shows post-progression survival with atezolizumab plus bevacizumab. The median survival time was 12.5 months (95% CI, 9.3 months—not reached).

The patients’ clinical data at the beginning of lenvatinib after atezolizumab plus bevacizumab are shown in Table 3. The Child–Pugh grades were A and B in 18 (75%) and 6 (25%) patients, respectively. The median time from the last atezolizumab plus bevacizumab to lenvatinib administration was 17 days.

Figure 4 shows the OS and PFS of lenvatinib after progressive disease of atezolizumab plus bevacizumab. The MST was 15.3 months (95% CI, 10.5 months—not reached), and median PFS was 4.0 months (95% CI, 2.5–6.4 months).

### 3.3. Antitumor Response to Lenvatinib Administration Following Atezolizumab Plus Bevacizumab

Table 4 and Figure 5 show the radiological responses to lenvatinib assessed according to the RECISTs criteria version 1.1 and mRECISTs. In total, 8 (33.3%) patients achieved PR, 10 (41.7%) SD, and 5 (20.8%) developed PD, with an ORR of 33.3% and DCR of 75.0%. On the other hand, using the mRECISTs criteria, 1 (4.2%) patient achieved CR, 12 (50.0%) patients achieved PR, 6 (25.0%) patients had SD, and 4 (16.6%) patients developed PD, with an ORR of 54.2% and DCR of 79.2%. It was not possible to evaluate RECISTs and mRECISTs in one patient because CT was not performed.

Regarding the tumor markers, the median AFP levels measured before and the first time after lenvatinib were 140.5 ng/mL and 70.6 ng/mL, respectively. The values decreased after lenvatinib, but the difference was not significant (*p* = 0.426).

### 3.4. Prognostic Factors for OS and PFS of Lenvatinib

Table 5 shows the prognostic factors for OS in patients on lenvatinib with progressive disease after first-line atezolizumab plus bevacizumab. In the univariate analysis, factors contributing to OS in patients on lenvatinib were the Child–Pugh class and the intrahepatic tumor occupancy rate. In the multivariate analysis, the following were independent factors contributing to OS in patients on lenvatinib: Child–Pugh class A (hazard ratio [HR], 0.14; 95% CI, 0.02–0.76; *p* = 0.02) and intrahepatic tumor occupancy rate less than 50% (HR, 0.03; 95% CI, 0.003–0.35; *p* < 0.01).

OS of patients on lenvatinib by Child–Pugh class was 17.5 months with class A and 10.5 months with class B, and OS by the intrahepatic tumor occupancy rate was 17.5 months with tumor occupancy less than 50% and 5.6 months with tumor occupancy greater than or equal to 50% (Figure 6).

On the other hand, there were no significant prognostic factors for the PFS of patients on lenvatinib in the univariate and multivariate analyses.

### 3.5. Treatment-Related Toxicities

Table 6 and Figure 7 show the AEs of lenvatinib after atezolizumab plus bevacizumab. AEs were observed in all patients. The most common AE was hypertension (15 patients, 62.5%), which was easily treated with antihypertensive therapy. Fatigue was the second most common AE (14 patients, 58.3%). Anorexia (45.8%), diarrhea (41.7%), proteinuria (29.2%), and hand–foot syndrome (29.2%) were also observed. Although there were some patients who had grade 3 or 4 AEs, no patients died because of AEs.

## 4. Discussion

In the present study, the efficacy of lenvatinib as second-line therapy after atezolizumab plus bevacizumab for patients with unresectable HCC was examined. The American Society of Clinical Oncology (ASCO) guidelines state that second-line treatment with TKIs, such as sorafenib, lenvatinib, cabozantinib, and regorafenib, may be feasible after first-line atezolizumab plus bevacizumab therapy, but the level of evidence for such treatment is low [12]. The ASCO guidelines state that the choice of second-line therapy should be based on patient and clinician preference, comorbidities, general condition, and the benefit of therapy. The European Society for Medical Oncology clinical practice guidelines recommend sorafenib, lenvatinib, regorafenib, cabozantinib, and ramucirumab side by side as second-line therapy after atezolizumab plus bevacizumab combination therapy [13]. A flowchart prepared by the American Association for the Study of Liver Diseases shows atezolizumab plus bevacizumab as first-line therapy, sorafenib and lenvatinib as second-line therapy, and cabozantinib, regorafenib, and ramucirumab as third-line therapy [14]. The Japan Society of Hepatology published clinical practice guidelines for hepatocellular carcinoma (fourth JSH-HCC guidelines) that listed atezolizumab plus bevacizumab as first-line therapy, lenvatinib and sorafenib as second-line therapies, and cabozantinib, regorafenib, and ramucirumab as third-line therapies. Furthermore, the 2021 edition of the JSH-HCC guidelines lists the combination of atezolizumab plus bevacizumab as first-line therapy, with lenvatinib, sorafenib, regorafenib, cabozantinib, and ramucirumab in parallel as second-line therapies. All guidelines have in common the recommendation of sorafenib or lenvatinib as first-line therapy and regorafenib, cabozantinib, or ramucirumab as second-line or later therapy when the combination of atezolizumab plus bevacizumab is difficult to use due to autoimmune disease or other reasons.

Recently, network analyses of OS and PFS have been reported for various molecular-targeted agents (MTAs) and immune checkpoint inhibitors (ICIs) for unresectable HCC by first-line and second-line treatments [15,16]. However, there have been no randomized, controlled trials in patients after atezolizumab plus bevacizumab, and given that the REFLECT trial comparing sorafenib and lenvatinib was a non-inferiority trial, it is considered that sorafenib, lenvatinib, regorafenib, cabozantinib, and ramucirumab in patients with AFP > 400 ng/mL are all candidate second-line therapies.

Despite the absence of established evidence for second-line therapy after first-line atezolizumab plus bevacizumab combination therapy for unresectable HCC, the results of the present study showed that subsequent systemic therapy for patients who progressed on first-line atezolizumab–bevacizumab was both safe and efficacious, with a response rate to lenvatinib of 33.3% by RECISTs and 54.2% by mRECISTs in 24 patients. Although PFS was 4.0 months, OS after atezolizumab plus bevacizumab was 15.3 months. In the REFLECT trial, the ORR of lenvatinib by mRECISTs was 40.6%, and the median OS was 13.6 months; thus, the present results showed better outcomes. In the multivariate analysis, patients with good liver function before lenvatinib after atezolizumab plus bevacizumab and those with a low intrahepatic tumor volume showed significant differences, contributing to the good therapeutic effect of lenvatinib after atezolizumab plus bevacizumab. Therefore, it is important to be careful not to decrease liver function with atezolizumab plus bevacizumab. Even when lenvatinib was used in first-line therapies, patients with better liver function were reported to have a better prognosis [17]. In addition, the therapeutic effect of lenvatinib after atezolizumab plus bevacizumab was found to be good when the tumor occupied a small percentage of the liver. Regarding AEs, hypertension, fatigue, anorexia, diarrhea, proteinuria, and hand–foot syndrome were seen in the present study, and they were similar to the side effects seen in the REFLECT study. Therefore, it was found that the AEs of lenvatinib after atezolizumab plus bevacizumab were comparable to those of lenvatinib alone.

One of the reasons why the combination of TKIs, such as lenvatinib, and ICIs, such as atezolizumab plus bevacizumab, is effective for HCC is due to the VEGF inhibition. HCC is a hypervascular tumor and is known to express high levels of VEGF [18]. In addition, VEGF is known to be associated with immunosuppression in the tumor microenvironment during the cancer immune cycle, creating an environment in which CD8-positive (CD8+) T cells are ineffective [19,20]. Specifically, VEGF suppresses dendritic cell maturation and T cell activation in lymph nodes, inhibits CD8+ T cell infiltration into tumors, and differentiates and induces immunosuppressive cells such as regulatory T cells (Tregs). VEGF inhibition may release these factors, thereby creating an environment in which CD8+ T cells can exert their effects.

The abundant infiltration of CD8+ T cells in tumor tissue is considered important for ICIs to have a good therapeutic effect [21,22]. It has also been reported that ICIs are less effective when there are many Tregs present that suppress the function of effector T cells such as CD8+ T cells [23,24]. To increase the efficacy of ICIs, it is important to balance the tumor microenvironment with a predominance of CD8+ cells over Tregs by inhibiting VEGF, as bevacizumab does. Similar to bevacizumab, lenvatinib is a multikinase inhibitor that targets not only VEGF receptors 1–3 but also fibroblast growth factor receptors 1–4, platelet-derived growth factor receptor alpha, rearranged during transfection, and KIT [25,26,27]. Therefore, the therapeutic effect of atezolizumab is expected to be enhanced by the VEGF inhibitory effect of lenvatinib.

It has been reported that a method to monitor the binding of anti-PD-1 antibodies to CD8+ T cells was found, and anti-PD-1 antibodies remain bound to CD8+ T cells for more than 20 weeks after patients stop treatment [28]. Therefore, it is assumed that the therapeutic effect of lenvatinib was added to the prolonged effect of the ICIs, and that it exerted its effect on immune cells and tumor cells.

Another reason why lenvatinib showed therapeutic efficacy after atezolizumab plus bevacizumab is that lenvatinib worked for ICI-refractory HCC. Factors that contribute to ICI refractoriness include Wnt/β-catenin signaling. It has been reported that the ICI treatment of HCC patients was associated with lower DCR, shorter median PFS, and shorter median OS if they had altered the activation of Wnt/β-catenin signaling [29]. On the other hand, it has been reported that Wnt/β-catenin mutations correlated with a high expression of FGFR4 and lenvatinib were shown to be highly effective against HCC with a high expression of FGFR4, with an ORR of 81% and PFS of 5.5 months, compared to the ORR of 31% and PFS of 2.5 months with a low expression of FGFR4 [30,31]. There was a case report that described a patient with HCC with a β-catenin mutation in whom lenvatinib after atezolizumab plus bevacizumab allowed the patient to progress to conversion surgery [32]. In addition, a case was reported in which a response to atezolizumab plus bevacizumab was achieved in the main intrahepatic tumor, but adrenal metastasis showed disease progression, and the subsequent administration of lenvatinib resulted in shrinkage of the adrenal metastasis and conversion to surgery [33]. There have been reports of differences in responses to ICIs by organ [34], including other carcinomas [35], and future analysis of the efficacy of second-line therapy in patients with the progression of extrahepatic lesions after ICIs as first-line therapy is also warranted.

Therefore, if the therapeutic effect of atezolizumab plus bevacizumab is judged to be poor, it may be better to change to lenvatinib immediately. As a way to examine the effects of atezolizumab plus bevacizumab at an early stage, there is a report that decreased AFP in unresectable HCC patients treated with atezolizumab plus bevacizumab at 3 weeks was identified as a factor predicting early tumor response [36]. This would help determine whether to continue atezolizumab plus bevacizumab at an early stage. In general, such tumor markers can be used to evaluate and predict the efficacy of treatment for HCC. Research on various biomarkers that can predict the efficacy of systemic therapy in HCC was also reported [37,38], and future studies are expected in the field of HCC, where systemic therapy is advancing.

Recently, the results of the phase III HIMALAYA trial of single tremelimumab and regular interval durvalumab (STRIDE) as standard first-line systemic therapy for unresectable HCC were published. STRIDE showed significantly better efficacy compared with sorafenib [39]. Then, we have to consider whether to use STRIDE for the second-line treatment of patients who received atezolizumab plus bevacizumab as first-line treatment. However, in other cancer types, in the treatment of renal cell carcinoma patients with disease progression during or after treatment with ICIs, combination therapy with atezolizumab and cabozantinib as a TKI did not improve PFS or OS compared with cabozantinib alone and was associated with an increase in serious AEs [40]. Considering AEs, this may be a factor in the choice of treatment with lenvatinib and other TKIs as second-line therapy after atezolizumab plus bevacizumab.

The present study has two limitations. First, it was retrospective. Second, a selection bias could have existed because of the clinical observational nature of this study. Nevertheless, the usefulness of lenvatinib after atezolizumab plus bevacizumab was demonstrated. We believe that this information will be very useful for the systemic therapy of patients with unresectable HCC. The further accumulation of evidence is expected to establish an effective treatment regimen after atezolizumab plus bevacizumab combination therapy.

## 5. Conclusions

Lenvatinib after atezolizumab plus bevacizumab for unresectable HCC is a safe treatment and should be considered as an effective option that may provide a good prognosis for patients with good liver function and low intrahepatic tumor volume.

## Figures and Tables

**Figure 1 cancers-15-05406-f001:**
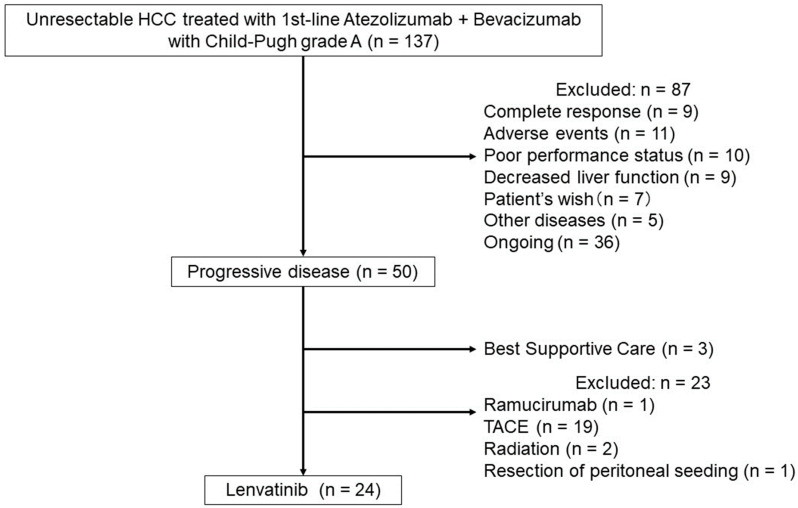
Flow chart of patient selection in the study.

**Figure 2 cancers-15-05406-f002:**
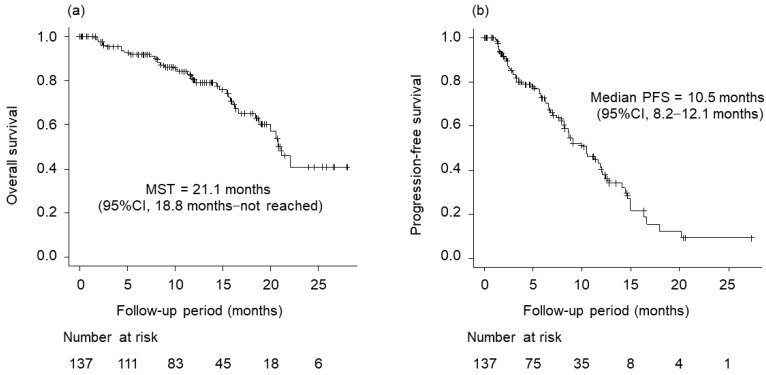
Overall survival (OS) and progression-free survival (PFS) of atezolizumab plus bevacizumab. (**a**) OS (median survival time (MST), 21.1 months; 95% confidence interval (CI), 18.8 months–not reached). (**b**) PFS (median PFS, 10.5 months; 95% CI, 8.2–12.1 months).

**Figure 3 cancers-15-05406-f003:**
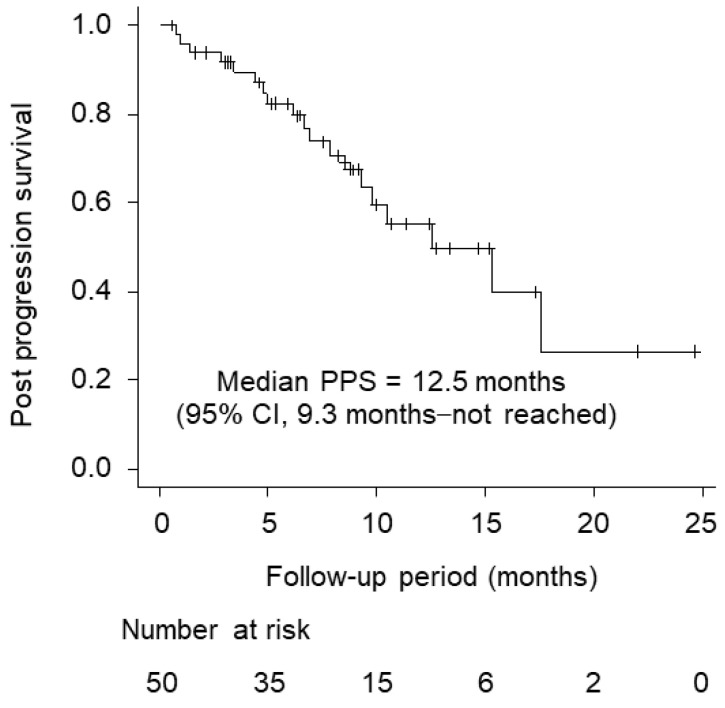
Post-progression survival (PPS) with atezolizumab plus bevacizumab (median PPS, 12.5 months; 95% CI, 9.3–12.5 months).

**Figure 4 cancers-15-05406-f004:**
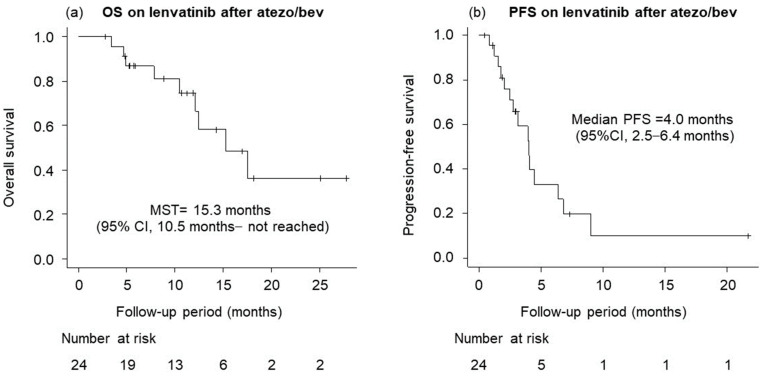
Overall survival (OS) and progression-free survival (PFS) of lenvatinib after atezolizumab plus bevacizumab. (**a**) OS (median survival time (MST), 15.3 months; 95% confidence interval (CI), 10.5 months–not reached). (**b**) PFS (median PFS, 4.0 months; 95% CI, 2.5–6.4 months).

**Figure 5 cancers-15-05406-f005:**
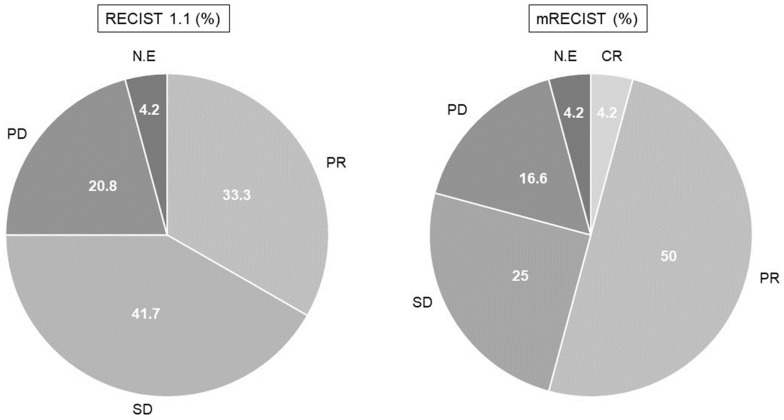
Radiological response (*n* = 24).

**Figure 6 cancers-15-05406-f006:**
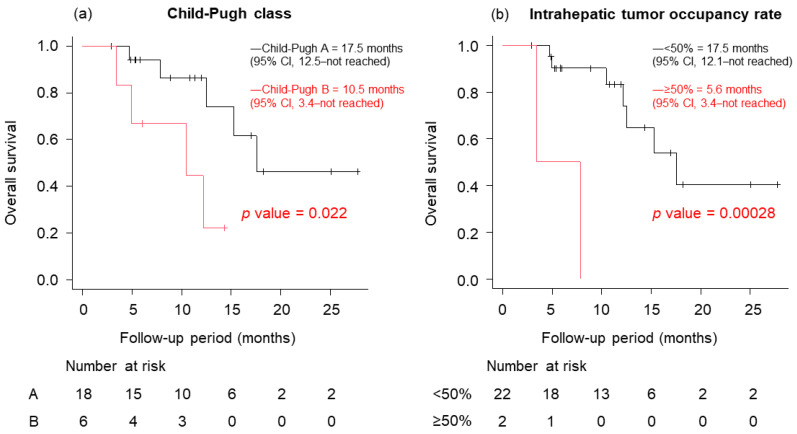
Overall survival (OS) of patients on lenvatinib after atezolizumab plus bevacizumab. (**a**) OS for patients with Child–Pugh A is better than for those with Child–Pugh B (median OS 17.5 vs. 10.5 months, *p* = 0.022). (**b**) OS for patients with a relative tumor volume (RTV) <50% is better than for those with an RTV ≥50% (median OS 17.5 vs. 5.6 months, *p* = 0.00028).

**Figure 7 cancers-15-05406-f007:**
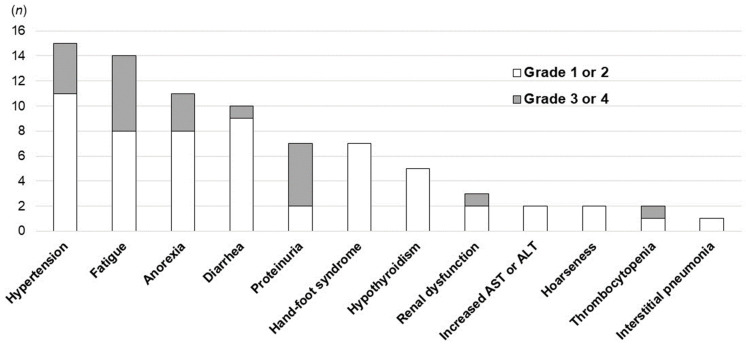
Adverse events associated with lenvatinib.

**Table 1 cancers-15-05406-t001:** Background characteristics of all cases (*n* = 137).

	All (*n* = 137)
Age, range, *y*	75 (47–92)
Sex (male/female), *n*	107/30
Performance status (0/1/2), *n*	113/18/6
Etiology (HBV/HCV/HBV+HCV/NBNC), *n*	15/48/2/72
Child–Pugh score (5/6), *n*	84/53
Relative tumor volume (<50%/≥50%), *n*	130/7
Size of main tumor, range, mm	28 (0–220)
Microvascular invasion (absent/present), *n*	112/25
Extrahepatic metastasis (absent/present), *n*	93/44
HCC stage (II/III/IVa/IVb), *n*	33/50/22/32
BCLC stage (A/B/C), *n*	8/63/66
Serum alpha-fetoprotein, range, ng/mL	18.1 (1.2–63642)
Serum des-gamma-carboxy prothrombin, range, mAU/mL	236 (11–197680)
Observation period, range, months	11.7 (1–28)

Values represent medians (range) or numbers of patients. HBV—hepatitis B virus infection; HCV—hepatitis C virus infection; NBNC—non-B, non-C viral hepatitis; HCC—hepatocellular carcinoma; BCLC—Barcelona Clinic Liver Cancer.

**Table 2 cancers-15-05406-t002:** Clinical data at the time of diagnosis of progressive disease regarding atezolizumab plus bevacizumab (*n* = 50).

	*n* = 50
Sex (male/female), *n*	41/9
Performance status (0/1/2), *n*	32/12/5
Etiology (HBV/HCV/NBNC), *n*	7/19/24
Child–Pugh grade (A/B/C), *n*	34/14/2
Relative tumor volume (<50%/≥50%), *n*	45/5
Size of main tumor, range, mm	33 (0–170)
Microvascular invasion (absent/present), *n*	35/15
Extrahepatic metastasis (absent/present), *n*	27/23
HCC stage (II/III/IVa/IVb), *n*	6/16/9/19
BCLC stage (A/B/C), *n*	3/15/32
Serum alpha-fetoprotein, range, ng/mL	105.3 (0.8–64620)
Serum des-gamma-carboxy prothrombin, range, mAU/mL	2842.5 (22–247805)
Systemic therapy after atezolizumab plus bevacizumab	25 (LEN 24, RAM 1)

Values represent medians (range) or numbers of patients. HBV—hepatitis B virus infection; HCV—hepatitis C virus infection; NBNC—non-B, non-C viral hepatitis; HCC—hepatocellular carcinoma; BCLC—Barcelona Clinic Liver Cancer; LEN—lenvatinib; RAM—ramucirumab.

**Table 3 cancers-15-05406-t003:** Clinical data at the beginning of lenvatinib after atezolizumab plus bevacizumab (*n* = 24).

	*n* = 24
Sex (male/female), *n*	20/4/
Performance status (0/1/2), *n*	19/4/1
Etiology (HBV/HCV/NBNC), *n*	5/11/8
Child–Pugh grade (A/B/C), *n*	34/14/2
Relative tumor volume (<50%/≥50%), *n*	22/2
Size of main tumor, range, *mm*	30 (0–120)
Microvascular invasion (absent/present), *n*	16/8
Extrahepatic metastasis (absent/present), *n*	13/11
HCC stage (II/III/IVa/IVb), *n*	2/9/3/10
BCLC stage (A/B/C), *n*	8/16
Serum alpha-fetoprotein, range, ng/mL	140.5 (1.5–64620)
Serum des-gamma-carboxy prothrombin, range, mAU/mL	2614 (25–214866)
Time to LEN administration after Atez+Bev, days	17 (4–63)

Values represent medians (range) or numbers of patients. HBV—hepatitis B virus infection; HCV—hepatitis C virus infection; NBNC—non-B, non-C viral hepatitis; HCC—hepatocellular carcinoma; BCLC—Barcelona Clinic Liver Cancer; LEN—lenvatinib; Atez+Bev—atezolizumab plus bevacizumab.

**Table 4 cancers-15-05406-t004:** Radiological response (*n* = 24).

%(*n*)	RECISTs 1.1	mRECISTs
CR	0.0 (0)	4.2 (1)
PR	33.3 (8)	50.0 (12)
SD	41.7 (10)	25.0 (6)
PD	20.8 (5)	16.6 (4)
N.E	4.2 (1)	4.2 (1)
ORR	33.3 (8)	54.2 (13)
DCR	75.0 (18)	79.2 (19)

RECISTs—response evaluation criteria in solid tumors; mRECISTs—modified response evaluation criteria in solid tumors; CR—complete response; PR—partial response; SD—stable disease; PD—progressive disease; N.E—not evaluated; ORR—objective response rate; DCR—disease control rate.

**Table 5 cancers-15-05406-t005:** Prognostic factors for overall survival in patients on lenvatinib with progressive disease after first-line atezolizumab plus bevacizumab (Cox hazards analysis).

Variable		Univariate	Multivariate		
		*p* Value	HR	95% CI	*p* Value
Sex	Male	0.13			
Etiology	Viral infection	0.72			
Child–Pugh grade	A	0.02	0.14	0.02–0.76	0.02
Microvascular invasion	Absent	0.63			
Extrahepatic metastasis	Absent	0.37			
Relative tumor volume	<50%	<0.01	0.03	0.003–0.35	<0.01
Serum alpha-fetoprotein	<400 ng/mL	0.40			
TACE/TAI combination	yes	0.13			

TACE—transcatheter arterial chemoembolization; TAI—transcatheter arterial infusion; HR—hazard ratio; CI—confidence interval.

**Table 6 cancers-15-05406-t006:** Adverse events associated with lenvatinib.

Event %(*n*)	All Patients (*n* = 24)	
	Any Grade	Grade 3 or 4
Hypertension	62.5 (15)	16.7 (4)
Fatigue	58.3 (14)	25.0 (6)
Anorexia	45.8 (11)	12.5 (3)
Diarrhea	41.7 (10)	4.2 (1)
Proteinuria	29.2 (7)	20.8 (5)
Hand–foot syndrome	29.2 (7)	0.0 (0)
Hypothyroidism	20.8 (5)	0.0 (0)
Renal dysfunction	12.5 (3)	4.2 (1)
Increased AST or ALT	8.3 (2)	0.0 (0)
Hoarseness	8.3 (2)	0.0 (0)
Thrombocytopenia	8.3 (2)	4.2 (1)
Interstitial pneumonia	4.2 (1)	0.0 (0)

AST—aspartate aminotransferase; ALT—alanine aminotransferase.

## Data Availability

The data that support the findings of this study are available from the corresponding author upon reasonable request.

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
