# Peer review of "Therapeutic Efficacy and Safety of Lenvatinib after Atezolizumab Plus Bevacizumab for Unresectable Hepatocellular Carcinoma"

_cancers, 2023, doi:10.3390/cancers15225406_

Round 1

Reviewer 1 Report

Comments and Suggestions for Authors

I'd like to commend the authors for their manuscript which I found well written and clear. 

Comments on the Quality of English Language

There are a few minor flaws.

Author Response

Thank you very much.

Reviewer 2 Report

Comments and Suggestions for Authors

The submitted manuscript entitled “Therapeutic efficacy and safety of lenvatinib after atezolizumab plus bevacizumab for unresectable hepatocellular carcinoma” focuses on the retrospective evaluation of the efficacy and safety of lenvatinib after atezolizumab plus bevacizumab for unresectable HCC. This study scientifically sounds and may be of interest for the journal audience. The manuscript is well-written and well-organized and it contains enough tables and figures to illustrate the results obtained in this study. However, there are some concerns and recommendations to improve the quality of the manuscript. There are as follows:

1.     In the Simple Summary, the phrase “lenvatinib…demonstrated good outcomes” – as the first- or second-line therapy? Please, clarify.

2.     In the Abstract, it is recommended to indicate what changes in biomarker levels were in response to the treatment.

3.     The Introduction is quite short. The authors used AFP and DCP levels, microvascular invasion, etc. for assessment of the treatment efficacy. Therefore, more discussion of these parameters is needed. Please, see and cite the following papers: doi: 10.3390/biomedicines9020159; doi: 10.5152/dir.2015.15125, etc.

4.     In the Materials and Methods, the authors should provide description of methods used to determine albumin-bilirubin grades as well as AFP and DCP levels.

5.     In the Results section:

a)     Table 1 contains background characteristics of all cases, and Table 2 contains characteristics at the time of the disease progression. There are no data on the efficacy of the combined treatment by atezolizumab and bevacizumab, i.e. after the combined treatment This should be given.

b)    Titles of subsections and each table contain the word “background”. On my opinion, Tables 2, 3, 4 do contain not background characteristics, but those after the combined treatments.

6.      English language grammar should be checked and corrected. For example, in the Simple Summary, it should be: “has been rapidly developed” and in section 3.1., it should be: “at the beginning”, etc.

Comments on the Quality of English Language

English is good

Reviewer 3 Report

Comments and Suggestions for Authors

The article “Therapeutic Efficacy and Safety of Lenvatinib after Atezolizumab plus Bevacizumab for Unresectable Hepatocellular Carcinoma” by Yano et al. is a retrospective study examining the efficacy and safety of lenvatinib after atezolizumab plus bevacizumab for unresectable HCC.

The article has the following shortcomings:

1.      In the abstract, on lines 39-40, the authors need to explain more rigorously how they included the 24 patients in the study.

2.      For the keywords: The authors did not adhere to MeSH for choosing keywords and have several imperfections in keyword formulations. There are keywords that are already in the title and therefore should NOT have been listed as keywords. Please, see line 50.

3.      The abstract is not written rigorously and credible, so it should be reformulated in the light of the protocol applied, results, and conclusions obtained.

4.      The postulation of some statements from the introduction based on the citation of only one article does not correspond to the scientific rigor related to Cancers.

5.      A better representation of the study and the work done is needed for the reader, to demonstrate the results.

6.      More comprehensive arguments and explanations of the results presented in the figures would be most welcome.

7.      Table 4. Radiological response should be better represented as a figure, i.e., a more suggestive diagram.

8.      Adverse events associated with lenvatinib presented in Table 6 should be better represented as a figure, i.e., a more suggestive diagram.

9.      More comparative arguments regarding the actual results obtained are missing. Discussions should better interpret the meaning of the results and explain why they matter.

For example:

See lines 293-297: please, rephrase more plausibly and convincingly. Please, also insert references.

See lines 308-311: please, rephrase more plausibly and convincingly. Please, also insert references.

10.   A “List of Abbreviations” must be completed and reviewed carefully and may be better presented in a table format at the end of the article.

I congratulate the authors for their work.

Overall, the study is interesting and deserves to be published with minor corrections. Overall, I recommend a minor revision.

I believe that after this minor revision provided by the authors on the issues suggested to be corrected and improved, it will provide useful and credible information for all readers, especially clinicians, and it is up to the Academic Editor to decide on its publication.

Thank you very much!

Reviewer 4 Report

Comments and Suggestions for Authors

This study depicts the course of 24 patients with palliative second-lline therapy of lenvatinib after Ate/beva. 

I have a few remarks

1. Introduction:  More focus on Lenva on why it is the candidate the authors chose to analyze

2. Methods: Please add percentages when reporting numbers

3. Results:

-Overall, only 24 patients were included for the focus of the study. Reporting of the 113 that only received ate/beva shifts the focus and does not add information. I suggest to remove the data.

-Please add percentages after numbers

-it might be intersting to have a comparsion (e.g. a group of patients receiving different second-line therapy. Maybe the authors can identify a subgroup wihtin the 113 patients that did not undergo lenva? In this case, the overall cohort shoud be presented.

-Fig4: Please add OS/PFS on lenva after ate/beva in title

Discussion_ The authors mention a predeicitve marker of ate/beva reponse by AFP decrease. Did they find something similar in their cohort?

Comments on the Quality of English Language

English needs to be improved with focus on expression and coherence.

Round 2

Reviewer 2 Report

Comments and Suggestions for Authors

In the revised manuscript, the authors have addressed the majority of my concerns. However, two of them remained non-addressed.

1.     Table 2 contains characteristics at the time of the disease progression. There are no data on the efficacy of the combined treatment by atezolizumab and bevacizumab, i.e. after the combined treatment This should be given.

The author’s” response: I am really sorry but I couldn’t understand the meaning. 

Here is explanation of my concern: In the Materials and Methods section. The authors stated that of total 137 patient “Thirty-six (26.3%) patients continued atezolizumab plus bevacizumab”. However, there only data on the efficacy of treatment with Lenvatinib after the combined treatment with atezolizumab plus bevacizumab. How was the efficacy and outcomes of those 36 patients? 

2.     The Introduction is quite short. The authors used AFP and DCP levels, microvascular invasion, etc. for assessment of the treatment efficacy. Therefore, more discussion of these parameters is needed

The author’s response: I added the sentence and references in line 61-64.

My remaining concern is that the manuscript focuses on the assessment of treatment response. Therefore, it is necessary to discuss the roles of predictive roles of biomarkers, i.e. their utility in giving information about treatment efficacy. This can be done in the Introduction or in the Discussion sections. Please, read, use for discussion and cite the following papers:  doi: 10.1080/14737159.2021.1987217 and doi: 10.3390/ijms24087640.

Comments on the Quality of English Language

English is quite good.

Reviewer 4 Report

Comments and Suggestions for Authors

I have no additional remarks

Comments on the Quality of English Language

mostly fine

Author Response

Thank you very much.